# Nomograms in Urologic Oncology: Lights and Shadows

**DOI:** 10.3390/jcm10050980

**Published:** 2021-03-02

**Authors:** Alessandro Morlacco, Daniele Modonutti, Giovanni Motterle, Francesca Martino, Fabrizio Dal Moro, Giacomo Novara

**Affiliations:** 1Urology Unit, Department of Surgical, Oncological and Gastroenterological Sciences, University of Padua, 35128 Padua, Italy; alessandro.morlacco@unipd.it (A.M.); modonutti.daniele@yahoo.it (D.M.); gio.motterle@gmail.com (G.M.); fabrizio.dalmoro@unipd.it (F.D.M.); 2Department of Nephrology, Dialysis and Kidney Transplant, International Renal Research Institute, San Bortolo Hospital, 36100 Vicenza, Italy; Francesca.martino.k@gmail.com

**Keywords:** nomogram, prostate cancer, bladder cancer, kidney cancer, renal function

## Abstract

Decision-making in urologic oncology involves integrating multiple clinical data to provide an answer to the needs of a single patient. Although the practice of medicine has always been an “art” involving experience, clinical data, scientific evidence and judgment, the creation of specialties and subspecialties has multiplied the challenges faced every day by physicians. In the last decades, with the field of urologic oncology becoming more and more complex, there has been a rise in tools capable of compounding several pieces of information and supporting clinical judgment and experience when approaching a difficult decision. The vast majority of these tools provide a risk of a certain event based on various information integrated in a mathematical model. Specifically, most decision-making tools in the field of urologic focus on the preoperative or postoperative phase and provide a prognostic or predictive risk assessment based on the available clinical and pathological data. More recently, imaging and genomic features started to be incorporated in these models in order to improve their accuracy. Genomic classifiers, look-up tables, regression trees, risk-stratification tools and nomograms are all examples of this effort. Nomograms are by far the most frequently used in clinical practice, but are also among the most controversial of these tools. This critical, narrative review will focus on the use, diffusion and limitations of nomograms in the field of urologic oncology.

## 1. Introduction

A nomogram is a graphical calculating device, a two-dimensional diagram designed to allow the approximate graphical computation of a mathematical function or equation. The graphical presentation may be a number of rulers where variables are listed separately, with a number of points assigned to a given magnitude of the variable. Then, the score obtained by the sum of all the variables is matched to a scale of outcome. In another version, the formula is in a computer or smartphone-based calculator, where specific variables are entered, and the results of the nomogram are provided to the user [1].

The development of a nomogram is a multi-phased process. Firstly, a clinical population where the nomogram will apply and a relevant clinical question should be chosen, with a clear definition of which outcome(s) are expected to be predicted by the tool. In a second phase, the variables (covariates) are selected. This process is crucial: the choice of covariates will have a profound influence on the performance of the nomogram, and it should be based on clinical significance (derived from existing evidence) rather than on statistical significance alone. Moreover, a statistical model must be chosen: logistic regression analysis with binary outcome is the most commonly used mathematical formula for prediction modelling in urology, while for survival analysis Cox proportional hazards model is used most of the times to fit Kaplan–Meier survival curves.

Before a widespread clinical use, the authors assess the performance of the nomogram: validation, discrimination and calibration. Validation refers to testing the tool in different populations, possibly unrelated to the training/development population (external validation). When external validation is not possible, internal validation (i.e., on the same dataset used for development) might still be acceptable but requires improvement using statistical methods; the more common are random extractions of different subsets of the database. On the other hand, calibration measures the precision of the estimate risk to predict the observed risk, better expressed in terms of agreement between predicted and observed probabilities. A calibration plot is commonly used to depict this aspect in a graphical form. Finally, discrimination is the ability of the nomogram to separate patients who will experience a certain outcome from those who will not, and it is usually expressed as an area under the curve (AUC) of a receiver-operating characteristic curve (ROC), also called concordance index (CI). The CI may vary from 0.5 (where the use of the nomogram is equal to throwing a coin—no better than chance) to 1 (the virtually impossible situation where the nomogram provides a perfect prediction in 100% of the cases).

Steyerberg et al. [2] have described a total of seven structured steps for the development of a prediction model: (1) consideration of the research question and initial data inspection; (2) coding of predictors; (3) model specification; (4) model estimation; (5) evaluation of model performance; (6) internal validation; and (7) model presentation. The model should then be validated using four key aspects: (1) calibration-in-the-large (the model intercept); (2) calibration slope; (3) discrimination, with a concordance statistic; and (4) clinical usefulness, involving a decision-curve analysis.

Ideally, to reach the highest possible level of evidence for adoption in clinical practice, every nomogram should prove its efficacy in a prospective, randomized clinical trial (RCT). The urological community, however, is very aware of the risk connected with a strict application of this principle. Most of the results discussed in the present review, in fact, come from good-level retrospective studies, which play an important role in providing timely and accurate clinical answers.

## 2. Prostate Cancer

Several moments in the diagnostic and therapeutic pathway of prostate cancer (pCa) may deserve a tool to assist in decision-making.

### 2.1. Diagnostic Phase

The first difficult question posed by a man with a clinical suspicion of pCa is “to biopsy or not”. Prostate biopsy is an invasive procedure related with risks of adverse events, such as hematuria, infection, worsening of urinary symptoms and even mortality. Furthermore, unnecessary prostate biopsies lead to over diagnosis of indolent pCa, with impact on quality of life and other health-related issues. For such reasons, several strategies have been developed in order to reduce the number of prostate biopsies identifying the men at higher risk of signification of pCa. Prostatic specific antigen (PSA) alone or free/total (F/T) PSA are widely used to stratify pre-biopsy pCa risk [3]. In order to improve the diagnostic performance of PSA, prostate health index (PHI) combines three forms of PSA: total PSA, free PSA and the isoform [−2]proPSA, and it can outperform total and free PSA for pCa detection on biopsy and have an association with aggressive forms of pCa [4,5,6]. PHI can also be combined with prostate volume to obtain PHI density, improving its diagnostic yield [7], or even with mpMRI with good results [8]. Since there is increasing evidence that pCa risk is multifactorial and not completely assessed by a single marker, prostate cancer risk calculators aim to estimate an individual’s risk for pCa based on multiple factors.

European Randomized Study of Screening for Prostate Cancer (ERSPC) calculator is available in different versions: two for lay people, where age, family history, urinary symptoms and PSA are taken into consideration; and two for physicians [9]. Another similar tool is the Prostate Cancer Prevention Trial (PCPT) risk calculator, which includes PSA, family history, digital rectal examination (DRE) and history of a prior negative biopsy. More recently, the calculator has been updated to include the results of urinary biomarkers such as pCa3 and MiPS, with improvement of diagnostic performance. However, a head-to-head comparison has shown that ERSPC outperforms PCPT in the prediction of any pCa and clinically-significant pCa. The Sunnybrook nomogram [10] combines age, urinary symptoms, PSA, free PSA, ethnic background, family history and digital rectal examination (DRE) to provide an estimate of pCa risk.

It is important to note that these calculators have not been assessed in RCTs, and their potential role in reducing pCa mortality remains unknown. For these reasons, all these calculators/nomograms are currently not recommended by National Comprehensive Cancer Network (NCCN) guidelines to decide whether prostate biopsy is indicated.

Nowadays these pre-biopsy tools are challenged by the growing role of upfront mpMRI. Van Leeuwen et al. [11] developed a nomogram integrating prostate MRI and clinical features to predict clinically significant pCa. The performance of the nomograms was greatly improved by inclusion of MRI results: application of the model would reduce 28% of prostate biopsies, while missing 2.6% of clinically significant pCa. However, the general applicability of the model has been questioned, especially because the range of PSA of the training population was quite narrow. Moreover, the multivariable model was not compared to MRI alone, it is therefore possible that most of the predictive value in this model is provided by MRI alone, which could in fact “obscure” the effect of the other variables.

More recently Bjurlin et al. showed that PSA density, age and MRI suspicion score can predict pCa on combined MRI-targeted and systematic biopsy and developed a nomogram including these data [12] with a ROC AUC for overall and clinically significant pCa detection of 0.78 and 0.84 for men without prior biopsy. Radtke et al. [13] developed a multivariable model on over 1100 men who underwent mpMRI prior to MRI/transrectal ultrasound fusion biopsy. The tool includes PSA, prostate volume, DRE and PI-RADS score as significant predictors of significant prostate cancer.

However, the role of MRI-based nomograms is not universally accepted: while the widespread use of upfront mpMRI is certainly gaining more and more space, the limitations of mpMRI (including the risk of missing clinically-significant pCa) must be carefully considered. As a matter of fact, the use of mpMRI as a component of a model and not as a single tool might indeed mitigate this risk. As always, the indication for prostate biopsy should be based on informed discussion with the patients and not merely on the results given by nomograms (with or without MRI).

### 2.2. Post-Diagnosis Decision Making

Another controversial setting is post-diagnosis decision making. An accurate risk stratification is very important in providing advice on the possible management of clinically localized pCa, given the different side effects associated with each treatment. This is particularly important when active surveillance (AS) is a viable option.

In 2008, Kattan et al. developed a statistical model to predict 120-month survival for pCa men not treated with curative intent. The variables included clinical stage, biopsy Gleason grade, method of diagnosis (TURP vs. biopsy), percent cancer, baseline PSA, age at diagnosis and the use hormonal therapy. Since then a number of predictive models have been developed based on clinicopathological variables [14,15]. Recently, Iremashvili et al. evaluated the ability of Kattan and Truong nomograms to select patients with Gleason 3 + 3 or 3 + 4 organ-confined pCa in a radical prostatectomy cohort, and compared it with that of AS criteria of John Hopkins (JH), University of California-San Francisco (UCSF) and the Prostate Cancer Research International: Active Surveillance (PRIAS) protocol. The results showed that nomograms were slightly more accurate than JH and UCSF but did not perform better than PRIAS criteria, which in turn demonstrated optimal balance between sensitivity and specificity in selecting patients with low-grade organ-confined pCa [16]. On the same issue, Davis et al. [17] pooled data on men suitable for AS but undergoing upfront radical prostatectomy and assessed the performance of four models in predicting non-clinically significant pCa (various definitions) and obtained AUCs from 0.618 to 0.664. These data suggest a moderate, non-completely convincing accuracy of these models.

The use of mpMRI-derived information is promising also in this setting. Siddiqui et al. developed and tested a model based on number of lesions, degree of suspicion at MRI and lesion volume/prostate volume to predict the probability of AS disqualification at subsequent biopsy. Defining a cutoff probability of 19% to 32% on the nomogram, the authors found that 27% to 68% of patients could avoid the biopsy. However, this model relies on a small cohort of rather selected patients and its applicability to the general population might be questioned.

Finally, new biomarker assays performed on prostate biopsies are providing additional insights on tumor biology and have shown promising results in risk reclassification during the initial decision-making. OncotypeDX^®^, Prolaris^®^, ProMark^®^ and Decipher have all been evaluated in this clinical setting [18], and the first two are now approved for clinical use by NCCN guidelines.

### 2.3. Before Primary Treatment

The evaluation of tumor extension and risk of residual/recurrent disease before primary treatment is definitely one of the most fertile areas for nomograms. The D’Amico risk classification was one of the first multifactorial models to stratify pCa according to adverse disease features and recurrence. Since then, many tools have been developed, mainly focusing on radical prostatectomy and aiming to predict lymph node invasion, positive surgical margins and extracapsular extension at definitive pathology [19]. The issue of lymph nodes is particularly relevant because with an accurate definition of pN+ risk a significant proportion of patients could avoid lymph node dissection and the associated morbidity. Based on clinical and biopsy variables, several nomograms have been developed [20,21]. One of the most used, the Briganti nomogram [22,23], is based on PSA, clinical stage, primary and secondary biopsy Gleason grade and percentage of positive cores, showing 87.6% accuracy and suggesting avoiding LND when the calculated risk is less than 5%. In its more recent version, published by the same group, the percentage of positive cores was subdivided into two categories (percentage of higher and lower grade cores/total cores) and showed a 90.8% accuracy. Using a 7% cutoff, this model would allow sparing almost 70% of pelvic lymph node dissections (PLNDs) with a risk of missing N+ of 1.5%. Obviously, this new model lacks a wide external validation and therefore its applicability to other RP populations is questionable.

Some authors have recently suggested that mpMRI could have a role in risk sub stratification in the setting of Briganti’s calculated risk < 5%, based on the high accuracy of mpMRI for extracapsular extension (ECE), seminal vesicle invasion (SVI) or high-grade disease detection (which in turn are associated with an increased risk of N+) [24].

Other well-known pre-RP nomograms are the Partin Tables [25] and the CAPRA score [26], along with MSKCC, Cagiannos and Godoy nomograms [27,28].

Another frequent clinical question is how to select men with high-risk pCa who will benefit the most from RP as opposed to men who are best served with other approaches. To provide a definitive answer to this question would ultimately require a RCT rather than a nomogram, however a multi-institutional group developed a model specifically aimed to this issue [29], identifying 40% of patients with high-risk pCa who have a specimen-confined disease at RP and improving the preoperative selection of these men.

As in the other setting, clinically based nomograms are challenged by the widespread adoption of mpMRI for staging purposes. Several reports have shown that mpMRI-derived information may have an incremental role when compared to clinical nomograms, although with some controversies [30,31,32,33]. In particular, almost all the studies show a significant performance of mpMRI (and improvement of nomograms accuracy) for local staging (extracapsular extension, seminal vesicle invasion), while the incremental role of mpMRI for nodal staging is limited (thus confirming the sub-optimal accuracy of mpMRI along with the good performance of nomograms in regards of this outcome). Martini et al. recently published on a mpMRI-based nomogram predicting side-specific extracapsular extension of prostate cancer on a model including PSA; highest ipsilateral biopsy Gleason grade; highest ipsilateral percentage core involvement; and extracapsular invasion on mpMRI. After internal validation, the model AUC was 82.11%, with excellent calibration especially when compared with mpMRI prediction of ECE alone. This model has been externally validated, with an AUC of 67.6%, however, the model showed a suboptimal calibration and the incremental value of adding mpMRI results to the other clinical variables was not statistically significant [34].

Several models based on similar integration are being presented in these years [35,36], although most of these nomograms lack an external validation and a formal comparison between these tolls has not been performed yet.

Gandaglia et al. developed a nomogram specifically aimed to predict lymph-node invasion (LNI) in MRI-diagnosed pCa. Briganti 2012, Briganti 2017 and MSKCC showed suboptimal performances in this subset; while a new model including PSA, cT stage, maximum diameter of the index lesion on mpMRI, grade group on MRI-guided biopsy and the presence of clinically significant pCa on concomitant systematic biopsy had an AUC of 86% [37]. This increased accuracy would translate into a higher number of unnecessary LND spared and lower risk of missing positive LNI compared to the existing models.

Finally, a novel nomogram to predict side-specific EPE has been recently proposed by Soeterik et al. The model includes PSA density, highest ipsilateral ISUP grade, side-specific percentage of positive cores on systematic biopsy and ipsilateral clinical stage assessed by both digital rectal examination and mpMRI. The use of mpMRI information significantly increased the AUC, while the model based on PSA density, ISUP grade and mpMRI stage was superior in terms of calibration [38].

When a head-to-head comparison between these nomograms was carried out, the Cagiannos model and the 2012-Briganti showed the best calibrations and results at the decision-curve analysis. On the other hand, the ability to avoid unnecessary lymph node dissection and the C-index values were virtually the same for all the nomograms tested in this study (Cagiannos, 2012-Briganti, Godoy and MSKCC) [39].

### 2.4. After Primary Treatment

After primary intervention, in particular when RP is performed, patients must be re-stratified to establish the need of additional treatments. This is a critical moment since post-RP treatments (in particular radiation therapy-RT-) carry a non-negligible risk of toxic effects.

Specifically, in a man with risk factors for recurrence after RP there are two main approaches: immediate adjuvant RT even in the setting of undetectable PSA or observation and early salvage RT if PSA starts to rise. Clearly, an accurate selection of patients more likely to need and benefit from RT and patients safely managed with observation would allow sparing a relevant proportion of men these side effects.

The CAPRA-S score [40] is a scoring system based on PSA, pathological Gleason Score (pGS), ECE and LNI (1 point each), positive surgical margins and SVI. Each point increase in CAPRA-S score carries a HR of 1.54 of pCa recurrence. The Stephenson and the Kattan nomograms [41,42] used similar variables obtaining good, but not optimal, accuracy. A common problem of these nomograms is that they rely largely on surgical/histopathological variables and most of them use composite definitions of pCa recurrence, including biochemical recurrence (BCR), clinical progression, need for salvage therapies and pCa mortality. Moreover, while they carry a good prognostic value, their predictive role in stratifying patients for additional therapies is more controversial.

To overcome these limitations, in the last 10 years several genomic-based biomarkers have been developed to predict both recurrence risk and adjuvant/salvage RT (ART/SRT) benefit. Of interest, Den et al. evaluated a cohort of 188 pCa patients at 10-year follow-up after ART or SRT for high-risk features. Genomic classifier score (Decipher^®^) outperformed conventional risk-assessment tools. Moreover, men with low genomic classifier (GC) scores could safely undergo salvage RT only in case of pCa recurrence, while patients with a high GC score are best served with ART [43]. The same tool was then recently integrated in a nomogram, which combines pathological variables such as pathological T stage (pT), Gleason score and lymph node status with the genomic classifier results. The results were provided as a sum of risk factors, including pT3b/T4 stage, GS 8-10, LNI and Decipher score > 0.6. Patients with two or more risk factors receiving ART had a decreased recurrence rate, while those with only one risk factor did not. This study indicates a possible way to exploit both clinical and genomic information further improving prediction models. Another interesting tool is PORTOS (Post-Operative Radiation Therapy Outcomes Score), a biomarker proteomic tool based on protein expression of a panel of 24 genes, which can predict individual response to RT after RP [44]. Prolaris^®^ and PTEN loss are other biomarkers used in this setting.

The high cost and the actual unavailability in many European countries are strong limitations to the widespread use of biomarkers. However, as evidence accumulates on the added benefit of these tools, it is possible that we will see an increase in clinical use of composite models (clinical and biomarkers) as an aid in post-prostatectomy decision-making.

## 3. Bladder Cancer

Diagnostic, predictive and therapeutic dilemmas are frequent also in bladder cancer (BC) management. Several tools and nomograms have been developed and apply to different settings of the disease. The most obvious clinical distinction is between the non-muscle invasive (NMIBC) setting and the muscle-invasive (MIBC) stage.

### 3.1. NMIBC: Prediction of Recurrence, Progression and Response to Treatment

When approaching a bladder tumor, the first step is almost invariably transurethral resection to evaluate the bladder status, obtain histology and eradicate the cancer whenever possible. Then, if NMIBC is confirmed, risk stratification is necessary to assess the need of additional treatments and optimize the follow-up schedule.

The European Association of Urology (EAU) guidelines stratify patients into three main groups: low, intermediate and high risk, but strongly recommend the use of two alternative models. The EORTC [45] scoring system is based on six clinical and pathological factors: number of tumors, diameter, previous recurrence, T stage, grade and concomitant carcinoma in situ (CIS). The sum will provide a recurrence score ranging 0–17 and a progression score ranging 0–23. The use of this system is limited by the exclusion of patients with CIS alone, the fact that no second-look TUR was performed in most patients and the very low rates of BCG (7%). A more recent version of the EORTC system overcomes the latter limitation and applies to patients treated with BCG induction and maintenance [46].

The Club Urológico Espanol de Tratamiento Oncológico (CUETO) provided another tool based on different trials evaluating BCG treatments [47].

Notwithstanding these limitations, the EORTC and CUETO systems are externally validated [48,49].

In 2005, a bladder cancer nomogram [50] was designed to predict the risk of recurrence and progression in NMIBC patients based on a multi-institutional cohort of nearly 2681 patients, including also a biomarker (nuclear matrix protein) and showing an accuracy of at least 84% for predicting BC recurrence. The main aim of this nomogram was to optimize follow-up schedules.

A 2015 review by Kluth et al. [51] listed six scores in addition to EORTC and CUETO and five NMIBC nomograms. Unfortunately, only one of these nomograms received external validation and applies to the Japanese population [52].

Since a considerable number of biomarkers have shown prognostic relevance in BC, their incorporation into nomograms is seen as the next frontier for improving tools and nomograms [53].

CDNA microarray multi-gene classifiers were developed by Dyrskjøt et al. [54] to predict pT stage, CIS and progression. Fristrup et al. [55] analyzed Ta/T1 tumors of genetically-different populations (Danish, Swedish, Spanish and Taiwanese) and defined four pivotal markers (TRIM29, UBE2C cyclin D1 and MCM7) acting as independent predictors of NMIBC progression.

On the other hand, Lindgren et al. [56] analyzed cDNA array to identify two intrinsic molecular subtypes differing in terms of gene expression, mutation profiles, level of genomic instability as well as in BC-free survival. Sjödahl et al. [57] analyzed 307 advanced bladder cancers and found 28 markers to classify urothelial carcinoma into urothelial-like, genomically unstable, basal/SCC-like, mesenchymal-like and small-cell/neuroendocrine-like. This further enhanced the complexity of BC molecular studies suggesting a systematic disagreement between mRNA profiling and by immunohistochemical profiling and therefore highlighting the need of a combination of these techniques to provide adequate classification.

High-risk NMIBC sub-stratification in often critical to identify potential candidates to early cystectomy due to the progression risk with conservative management. Van Kessel et al. [58] investigated the incremental value of biomarkers to improve NMIBC risk stratification with regards to progression and studied multiple markers of methylation and mutation. These markers were then incorporated into an EORTC table. Taking into consideration the progression rate for high-risk patients according to EAU/EORTC (4.25/year/100 patients), a combination of FGFR3 mutation status and GATA2 methylation status was able to reclassify these patients into a good class (26.2% with a progression rate of 0.86), a moderate class (49.7% of patients, 4.32) and a poor class—very high risk (24.0%, 7.66).

From the clinical standpoint, it is not completely clear in this paper how many patients would have been in the ‘highest risk’ EAU subcategories and therefore the added benefit of biomarker in this specific setting could not be assessed. However, these biomarkers could be worth using also in intermediate or even low-risk categories, but this needs to be further explored and validated in other studies.

Finally, a number of studies focused on BCG response in link with genetic variations. As summarized in a recent systematic review [59], most works analyzed single variants on rather small groups of patients without validation. However, several pathways deserve attention: inflammatory genes (particularly polymorphisms in IL-6, IL-8 and TNF-alpha), glutathione pathway genes (GSS, GPX2), nucleotide excision repair (NER) genes, sonic hedgehog (Shh) genes and apoptosis genes. Since single biomarkers seem insufficient to explain the complexity of BCG antitumor effect, further research should evaluate combination of biomarkers in larger studies, and then a well-designed predictive model for clinical use could be built.

### 3.2. MIBC: Prediction of RC Outcomes and Long-Term Survival

As far as the cystectomy population is concerned, two main areas of interest are present: the prediction of outcomes at radical cystectomy in order to counsel patients and optimize access to additional (particularly neoadjuvant chemotherapy-NAC-) treatments and the long-term oncological prognosis after primary treatment, in order to select patients for adjuvant therapies.

The Karakiewicz nomogram, published in 2006, relies on T stage, 1973 WHO tumor grade, presence of carcinoma in situ, age, gender and delivery of neoadjuvant chemotherapy and showed a better performance than TUR stage alone to predict pT3-4 at RC [60]. Similarly, the Green’s model excludes NAC patients and predicts non-organ confined disease at RC using T stage, lymphovascular invasion (LVI) and radiographic evidence of non-organ confined disease or hydronephrosis [61]. Both tools received external validation, but the Green’s one is more recent and includes more contemporary variables (in particular CT/MRI features). The main use of this nomogram could be a better selection of NAC candidates, but this is not supported by the guidelines or strong clinical evidence at the present moment.

In 2015 Kluth et al. [51] summarized BC nomograms reporting six nomograms predicting adverse RC pathology with similar variables and accuracies ranging 68%–83%. Unfortunately, almost all lacked external validation and their clinical utility is questionable.

When disease recurrence and survival after RC are taken into consideration, the same review provides 28 publications. Most of these nomograms rely on clinical and pathological features (T and N stage, pathologic tumor grade, presence of LVI and CIS, administration of neoadjuvant or adjuvant chemotherapy and/or radiation therapy), providing RFS, CSS and seldom OS estimates with accuracies ranging 64%–87%. Also in this setting, many models are not externally validated (only nine out of 28). However, some of these studies deserve further attention.

A postoperative nomogram was developed and published by the International Bladder Cancer Nomogram Consortium (IBCNC) [62]. This tool helps to predict the five-year risk of disease recurrence based on age, time from diagnosis to surgery, gender, pathologic tumor stage, tumor grade, histologic type and N status. A discrimination (c-index) analysis showed a value of 0.78, which was compared to and outperformed the TNM alone or the pathologic group model.

Vickers et al. evaluated the same model using a decision curve analysis to establish whether the nomogram can improve decision making in choosing candidates to adjuvant chemotherapy after RC. Analyzing 4462 patients, the authors compared different cut-off levels provided by the nomogram (10%, 25% and 70% risk of five-year disease recurrence) and TNM pT/pN stage criteria and found that the use of the nomogram outperformed TNM stage for the indication to receive adjuvant chemotherapy in every scenario. In particular, the use of a nomogram-based decision resulted in 60 fewer chemotherapy treatments per 1000 patients without any increase in recurrence rates.

On the same page, the Bladder Cancer Research Consortium (BCRC) nomogram [63] is based on pT/pN status, tumor grade, presence of LVI and CIS at RC, and the use of chemotherapy (neoadjuvant, adjuvant or both) and/or radiation therapy. This model can predict the risk of disease recurrence, cancer-specific mortality and overall mortality at different time intervals with c-indexes of 0.78 for recurrence and CSM and 0.73 for OM.

As we have shown, most MIBC nomograms focus on RC outcomes. International guidelines currently recommend neoadjuvant chemotherapy (NAC) in all MIBC before RC. However, there are subsets of patients who clearly benefit from this treatment and others who do not or, even worse, progress during NAC. For these reasons, a tool predicting the response to NAC and its oncological impact would be really useful. Steps in this direction started several years ago: in 2005 Takata et al. [64] carried out a genome-wide study and identified 14 “predictive” genes showing the most significant differences between responders and non-responders. More recently, different molecular subtypes have been identified based on multi-gene expression (transcriptome analysis) and these subtypes (including claudin-low, basal, luminal-infiltrated and luminal tumors) had different prognostic features. In particular, basal tumors showed the most improvement in OS with NAC compared with surgery alone, while luminal tumors had the best survival regardless of NAC and claudin-low tumors had poor prognosis irrespective of treatment regimen, clearly setting the need for more effective therapies [65]. However, these biomarkers have not entered clinical practice, also because they have not yet been translated into easy-to-use clinical tools. Several trials are currently analyzing the impact of molecular subtyping in choosing individualized therapies and the results are eagerly awaited to provide answers to these important questions [66].

Despite their potential utility in this specific setting, however, nomograms and predictive models are still underused in the setting of MIBC, possibly because of heterogeneity in the therapeutic choices between different centers.

## 4. Kidney Cancer

Proper selection of patients is key in surgical treatment of patients with renal cell carcinoma (RCC). Consequently, several nomograms have focused on this issue, investigating either oncological or functional or surgical (i.e., complications) outcomes. Many models, in particular, aimed to predict complications, functional results and oncological outcomes of partial nephrectomy (PN) as compared to radical nephrectomy (RN), while others dealt with radical nephrectomy and its oncological implications.

The Kattan nomogram was developed to predict five-year recurrence after RN for RCC. The model had a c-accuracy of 0.74 [67]. Anyhow, this system was tested in different populations with controversial results. For example, a study on French patients found a c-index of 0.607 for relapse-free survival and on multivariate Cox analysis TMN stage was the only significant predictor of the outcome [68].

The Karakiewicz nomogram originated from a multi-institutional series and was externally validated to predict RCC-specific mortality after nephrectomy for all stages [69,70]. The model, based on age, gender, symptoms and TMN stage showed high accuracy with 88.1% accuracy at one year, 86.8% at five year and 84.2% at ten year.

In the specific setting of locally-advanced RC, two recent studies analyzed survival for surgically-treated RCC with tumor thrombus and developed predictive tools.

Abel et al. reported on 636 patients and found a role of tumor diameter, body mass index, low preoperative hemoglobin, thrombus level, perinephric fat invasion and non clear-cell histology in predicting five-year recurrence free survival. The overall accuracy of the model was good with a c-index of 0.72, better than other previous models such as the UISS (UCLA Integrated Staging System), SSIGN (Stage, Size, Grade and Necrosis) and Sorbellini model [71]. The model was not externally validated.

Gu et al. [72], in a similar study, developed a nomogram based on histological subtype, collecting system invasion, metastasis at surgery, De Ritis ratio (AST/ALT) and serum albumin. The model showed a c index of 0.75 for overall survival. However, almost 60% of the patients included had thrombus in the renal vein only and the study lacks external validation, therefore it has very limited application in the IVC thrombus population.

In the field of partial nephrectomy, many tools are available. The PADUA [73] and the RENAL [74] systems are based on anatomical features of the lesion and aim to predict the complication risk after partial nephrectomy and, with regards to the RENAL score, the malignant potential. These systems have been in use for several years and were externally validated. However, the predictive role of the RENAL score in terms of malignant histology is controversial: while some studies confirmed this value and provided external validation [75], other works did not find a significant predictive value especially in the small renal masses (where size is an obvious limiting factor) [76,77].

More recently, Karlo et al. proposed a model based on a nephrectomy database aiming to identify cortical indolent tumors based on clinical and CT scan features (including necrosis, calcification, contour, renal vein invasion, collecting system invasion, etc.) [78]. The model developed from this study had a c-index of 0.82. The authors should be commended for developing a statistical model based on these features, however the effort of defining the behavior of renal masses based only on clinical and imaging features has been around in the urology field for a long time with controversial results.

Finally, nomograms predicting the functional impact of PN have seen a development in the last years. Remarkably, Martini et al. [79] created a nomogram based on age, sex, Charlson comorbidity index, baseline eGFR, RENAL nephrometry score and the occurrence of acute kidney injury (AKI) in patients with normal baseline renal function or in patients with chronic kidney disease. This model shoved a c-index of 0.73 in evaluating the risk of significant eGFR reduction (<25% from baseline after 3 to 15 months after a robot-assisted partial nephrectomy). Notably, preoperative renal function and comorbidities played a major role in defining the risk of AKI while ischemia time (at a median, IQR of 15 min, 11–20 min) did not have a significant role as a predictor of acute kidney injury and was not included in the model, while only 7% of the patients underwent a clampless approach.

Similarly, Shum et al. [80] included a number of preoperative factors to build a nomogram with a c-accuracy of 0.61 and 0.7 (MDRD and CKD-EPI formulae) for the prediction of eGFR one year after partial nephrectomy. Although interesting in its aim, this model was not externally validated and showed overall a moderate accuracy.

## 5. Conclusions

Urologic oncology is certainly a fertile field for nomograms and predictive models. However, evidence of efficacy, availability (as web-based or apps) and ease of use will probably be the key factors for the success of a nomogram. The continued proliferation of “new” models lacking external evaluation and with unproven clinical benefit is not necessarily providing help in everyday decision-making. On the other hand, refining the existing validated models, possibly including new variables with a proven role (such as biomarkers or mpMRI in prostate cancer) could improve the accuracy of “old” nomograms.

As already noted by Catto more than ten years ago [81], rather than multiply the number of nomograms, we need better evidence of efficacy. There is a fundamental question about a predictive model: is this tool improving patient care and treatment outcomes? Perhaps the only way to provide a reliable answer is to promote prospective comparative trials where nomograms can show their efficacy in direct clinical care, improving outcomes and quality of life and controlling treatment costs.

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
