# Peer review of "Nomograms in Urologic Oncology: Lights and Shadows"

_jcm, 2021, doi:10.3390/jcm10050980_

Round 1

Reviewer 1 Report

This is an interesting, well written overview of the present literature regarding regularly used tools in urological practice: nomograms.

Please find below some suggestions to consider which could improve this interesting manuscript:

Introduction:

  • The authors mention the development of a nomogram as multi-phased process. Please consider articles of Steyerberg (doi: 10.1093/eurheartj/ehu207) and/or Moons and colleagues (doi: 10.7326/M14-0698.), in which a comprehensive and step-wise approach for the development of prediction tools is described. 
  • Please also mention the logistic regression analysis with binary outcome as the most commonly used mathematical formula for prediction modelling in urology (e.g. prediction of lymph node metastasis in patients undergoing ePLND)
  • Please provide adequate definitions for calibration: which includes the agreement between predicted and observed probabilities (calibration measures the precision of the estimate risk to 61 predict the observed risk: this sentence can be improved)

Chapter prostate cancer

  1. Diagnostic phase: please reflect on the benefits and potential disadvantages of using mpMRI in risk prediction tools to assess the risk of presence of prostate cancer. Do you recommend standard use of mpMRI based prediction models to determine indication for biopsy?
  2. Which risk calculator is the preferred one: the ERSPC calculator? If yes, which one?
  3. Have risk calculators which can be used to determine indication for biopsy been externally validated sufficiently yet? Or is there a need for external validation studies?

Before primary treatment (prostate cancer)

  1. Please elaborate which of the commonly used pre-RP nomograms have been externally validated; and if so which nomograms performed at best and should be recommended?
  2. Please consider referring to external validation studies performed of the Martini nomogram (Sighinolfi et al doi: 10.1111/bju.14665, and Soeterik et al. doi: 10.1016/j.urolonc.2019.12.028)
  3. Please also consider referring to a recently published study proposing another novel side-specific EPE nomogram including mpMRI features (doi: 10.1016/j.euo.2020.08.008)

Bladder cancer

  1. Bladder cancer nomograms are less frequently used in daily practice, and are also less recommended by the EAU guidelines. Could you provide a reason for that? Is risk prediction for bladder cancer more complex compared with prostate cancer?

General methodological considerations:

In case this is a narrative review: please clearly state this.

If some form of structured methodology has been used, please report on this (PRISMA Guidelines: inclusion criteria, study inclusion flow chart including title/abstract screening, full text selection etc)

Author Response

This is an interesting, well written overview of the present literature regarding regularly used tools in urological practice: nomograms.

Please find below some suggestions to consider which could improve this interesting manuscript:

Introduction:

The authors mention the development of a nomogram as multi-phased process. Please consider articles of Steyerberg (doi: 10.1093/eurheartj/ehu207) and/or Moons and colleagues (doi: 10.7326/M14-0698.), in which a comprehensive and step-wise approach for the development of prediction tools is described.

Thanks for this suggestion. A synthesis of the structured steps proposed by  Steyerberg et al has been inserted at lines 74-80.

Please also mention the logistic regression analysis with binary outcome as the most commonly used mathematical formula for prediction modelling in urology (e.g. prediction of lymph node metastasis in patients undergoing ePLND)

Thanks for this suggestion, a sentence has been added (ll 54-55)

Please provide adequate definitions for calibration: which includes the agreement between predicted and observed probabilities (calibration measures the precision of the estimate risk to 61 predict the observed risk: this sentence can be improved)

This has been added to the cited sentence to improve its clarity.

Chapter prostate cancer

Diagnostic phase: please reflect on the benefits and potential disadvantages of using mpMRI in risk prediction tools to assess the risk of presence of prostate cancer. Do you recommend standard use of mpMRI based prediction models to determine indication for biopsy?

We do agree with the reviewer that this aspect deserves further discussion, therefore a sentence has been added to this section: “However, the role of MRI-based nomograms is not universally accepted: while the widespread use of upfront mpMRI is certainly gaining more and more space, on the other hand not all patients are having mpMRI when the first biopsy is under consideration. Moreover, the limitations of mpMRI (including the risk of missing clinically-significant pCa) must be carefully considered [14]. As a matter of fact, the use of mpMRI as a component of a model and not as a single tool might indeed mitigate this risk.  As always, the indication for prostate biopsy should be based on  informed discussion with the patient and not merely on the results given by nomograms (with or without mpMRI).”

Which risk calculator is the preferred one: the ERSPC calculator? If yes, which one?

Have risk calculators which can be used to determine indication for biopsy been externally validated sufficiently yet? Or is there a need for external validation studies?

Thanks for raising this point. The authors feel that providing a definitive recommendation for the use of a single nomograms would go beyond the aim of the present review, however, the available evidence has been discussed (see for reference this paragraph “However, a head-to-head comparison has shown that ERSPC outperforms PCPT in the prediction of any pCa and clinically-significant pCa. The Sunnybrook nomogram [10] combines age, urinary symptoms, PSA, free PSA, ethnic background, family history and DRE to provide an estimate of pCa risk.

It is important to note that these calculators have not been assessed in RCTs, their potential role in reducing pCa mortality remain unknown.  For these reasons, all these calculators/nomograms are currently not recommended by NCCN guidelines to decide whether prostate biopsy is indicated.  ”)

Before primary treatment (prostate cancer)

Please elaborate which of the commonly used pre-RP nomograms have been externally validated; and if so which nomograms performed at best and should be recommended?

This is an interesting point. The results of a head-to-head comparison between four nomograms have been discussed in the text: “When a head-to-head comparison between these nomograms was carried out, the Cagiannos model and the 2012-Briganti showed the the best calibrations and results at the decision-curve analysis. On the other hand, the ability to avoid unnecessary lymph node dissection and the C-index values were virtually the same for all the nomograms tested in this study (Cagiannos, 2012-Briganti, Godoy and MSKCC) [37].”.

Please consider referring to external validation studies performed of the Martini nomogram (Sighinolfi et al doi: 10.1111/bju.14665, and Soeterik et al. doi: 10.1016/j.urolonc.2019.12.028)

Thanks, a note and a reference has been added.

Please also consider referring to a recently published study proposing another novel side-specific EPE nomogram including mpMRI features (doi: 10.1016/j.euo.2020.08.008)

Thanks for the suggestion. A brief description of this model has been provided (ll)

Bladder cancer

Bladder cancer nomograms are less frequently used in daily practice, and are also less recommended by the EAU guidelines. Could you provide a reason for that? Is risk prediction for bladder cancer more complex compared with prostate cancer?

Bladder cancer is more rare than prostate cancer, however, several predictive models are available both in NMIBC and in MIBC. The authors do not completely agree that EAU guidelines do not recommend the use of predictive models: as a matter of fact the guidelines strongly advise the use of EORTC or CUETO scores for risk stratification in NMIBC. However, we do agree that in MIBC nomograms are not routinely used in most institution, as stated in a brief comment at the end of the MIBC section “Despite their potential utility in this specific setting, however, nomograms and predictive models are still underused in the setting of MIBC, possibly because of heterogeneity in the therapeutic choices between different centers.

General methodological considerations:

In case this is a narrative review: please clearly state this.

This has been added to line 34

If some form of structured methodology has been used, please report on this (PRISMA Guidelines: inclusion criteria, study inclusion flow chart including title/abstract screening, full text selection etc)

This is a narrative review – so no structured methodology was adopted.

Reviewer 2 Report

This is an interesting, comprehensive and generally well-written review paper. Much of it is very technical, although I appreciate its usefulness to specialists in the area. Overall I have only a few comments intended to improve the paper as below.

Page 1, abstract – I thought you might put some brief summary of the levels of efficacy achieved by existing nomograms in your abstract, since it currently contains little quantitative information.

Page 2, line 76 – “infection, worsening of LUTS, etc.)” – its best not to list things in parentheses and then have “etc” on the end. Better to say “including…” and put a more definitive list. Who knows what you mean by “etc”. Also this sentence is the beginning of a plethora of undefined acronyms in coming pages (including PSA, LND, PLND, ERSPC, PCPT, PI-RAD, PCA3, NCCN, DRE, MiPS, and PRIAS). I realise many of these are common in your field, but my understanding is that all acronyms except SI Units should be defined on first use.

Page 3, line 98 – I thought it was interesting that the NCCN do not recommend nomograms because they are waiting for the definitive results of RCT trials – clearly they may be waiting a long time, while existing nomograms appear to have substantial potential to benefit patients. In my view this is an ill-informed approach, and given that you present cogent results from cohort studies (i.e. non-RCTs) elsewhere in your review, I thought you might comment on the mismatch between the available or possible evidence and the evidence expectations of committees – this might fit nicely in your conclusion. You might also want to consider a discussion of precisely this issue published recently: Webster CS. Evidence and efficacy: time to think beyond the traditional randomised controlled trial in patient safety studies. Br J Anaesth. 2019 Jun;122(6):723-725. doi: 10.1016/j.bja.2019.02.023.

Page 7, line 324 – “The Karakiewicz’ nomogram” – this doesn’t require an apostrophe of possession.

Page 10, line 438 – “therefore the variation in Although interesting” – there sentence appears to have a missing part.

END

Author Response

This is an interesting, comprehensive and generally well-written review paper. Much of it is very technical, although I appreciate its usefulness to specialists in the area. Overall I have only a few comments intended to improve the paper as below.

 Page 1, abstract – I thought you might put some brief summary of the levels of efficacy achieved by existing nomograms in your abstract, since it currently contains little quantitative information.

Thanks for this suggestion. The authors actually feel that, given the amount of different data discussed in the text and the space constraints in the abstract, inserting complete or even synthetic quantitative information would risk to alter its readability without providing a real help to the reader.

Page 2, line 76 – “infection, worsening of LUTS, etc.)” – its best not to list things in parentheses and then have “etc” on the end. Better to say “including…” and put a more definitive list. Who knows what you mean by “etc”. Also this sentence is the beginning of a plethora of undefined acronyms in coming pages (including PSA, LND, PLND, ERSPC, PCPT, PI-RAD, PCA3, NCCN, DRE, MiPS, and PRIAS). I realise many of these are common in your field, but my understanding is that all acronyms except SI Units should be defined on first use.

Thanks for this suggestion. The acronyms have been declared whenever possible, and the ‘etc’ corrected.

Page 3, line 98 – I thought it was interesting that the NCCN do not recommend nomograms because they are waiting for the definitive results of RCT trials – clearly they may be waiting a long time, while existing nomograms appear to have substantial potential to benefit patients. In my view this is an ill-informed approach, and given that you present cogent results from cohort studies (i.e. non-RCTs) elsewhere in your review, I thought you might comment on the mismatch between the available or possible evidence and the evidence expectations of committees – this might fit nicely in your conclusion. You might also want to consider a discussion of precisely this issue published recently: Webster CS. Evidence and efficacy: time to think beyond the traditional randomised controlled trial in patient safety studies. Br J Anaesth. 2019 Jun;122(6):723-725. doi: 10.1016/j.bja.2019.02.023.

We do agree with this principle, therefore a paragraph has been added to the initial discussion about nomograms “Ideally, to reach the highest possible level of evidence for adoption in clinical practice, every nomogram should prove its efficacy in a prospective, randomized clinical trial (RCT). The urological community, however, is very aware of the risk connected with a strict application of this principle. Most of the results discussed in the present review, in fact, come from good-level retrospective studies, which play an important role in providing timely and accurate clinical answers.”

Page 7, line 324 – “The Karakiewicz’ nomogram” – this doesn’t require an apostrophe of possession.

Page 10, line 438 – “therefore the variation in Although interesting” – there sentence appears to have a missing part.

Both these errors have been corrected, thanks.